# The Influence of HLA Alleles on the Affective Distress Profile

**DOI:** 10.3390/ijerph191912608

**Published:** 2022-10-02

**Authors:** Mihaela Laura Vică, Cristian Delcea, Gabriela Alina Dumitrel, Mihaela Elvira Vușcan, Horea Vladi Matei, Cosmin Adrian Teodoru, Costel Vasile Siserman

**Affiliations:** 1Department of Cellular and Molecular Biology, “Iuliu Hațieganu” University of Medicine and Pharmacy, 400012 Cluj-Napoca, Romania; 2Institute of Legal Medicine, 400006 Cluj-Napoca, Romania; 3Faculty of Psychology, “Tibiscus” University, 300559 Timișoara, Romania; 4Faculty of Medicine, “Iuliu Hațieganu” University of Medicine and Pharmacy, 400012 Cluj-Napoca, Romania; 5Faculty of Industrial Chemistry and Environmental Engineering, Polytechnic University, 300223 Timișoara, Romania; 6Clinical Surgical Department, Faculty of Medicine, “Lucian Blaga” University, 550002 Sibiu, Romania; 7Department of Legal Medicine, “Iuliu Hațieganu” University of Medicine and Pharmacy, 400012 Cluj-Napoca, Romania

**Keywords:** anxiety, affective distress, HLA, univariate analysis

## Abstract

(1) Background: Affective distress can be triggered by aggressive stimuli with an unfavorable role for the individual. Some of the factors that lead to the development and evolution of a mental disorder can be genetic. The aim of this study is to determine some correlations between the human leukocyte antigen (HLA) genes and the affective distress profile (PDA). (2) Methods: A psychological assessment and testing tool for anxiety was applied to 115 people. The low-resolution HLA alleles of class I (HLA-A, HLA-B, and HLA-C) and class II (HLA-DRB1 and HLA-DQB1) were identified by the PCR technique after DNA extraction from the blood. Depending on the PDA, the subjects were divided into two groups: a group with a low PDA and another one with a medium and high PDA. The IBM SPSS software was used to compare the frequency of HLA alleles between the two groups. (3) Results: The univariate analysis revealed a significant association of the HLA-A locus (A*01, A*30), HLA-B (B*08), and HLA-DRB1 (DRB1*11) with the low PDA group and of the HLA-A locus (A*32), HLA-B (B*52), and HLA-C (C*12) with the medium and high PDA group. (4) Conclusions: The present study highlighted potential associations between HLA alleles and anxiety disorders.

## 1. Introduction

Affective distress begins with the general concept of stress that can be triggered by aggressor stimuli with negative meaning, involving the functional and dysfunctional negative emotions associated with the states of anxiety and depression. Negative stressors have an unfavorable role for the individual, being considered, from the perspective of the cognitive and behavioral theory of the transactional model of stress, a cognitive and behavioral effort to reduce, control, or tolerate external demands that exceed personal resources [1].

The psychological perspective approaches the concept of distress as a relationship between the environment and the individual, which involves demands that exceed the individual’s own internal resources and threaten the inner balance, causing the triggering of coping mechanisms, cognitive, affective, and behavioral responses. Therefore, the stress factor represents an event or an external or internal situation, real or imaginary, which requires adaptive reactions from the individual [1]. Carrying out a paternity test can become a stressful factor, being associated as a determining event in the individual’s life, which can have significant repercussions on the future. The triggering of negative automatic thoughts causes negative emotions, and the individual feels a state of anxiety through the prism of insecurity, ignorance of the procedures for determining paternity, as well as the maladaptive coping mechanisms that can intervene.

In mental disorders, some studies have indicated that genetic factors are among the causes or precursors to the development of a mental illness. At the same time, in clinical practice, patients often complain that change “is impossible, because their disorder depends on the genes.” For example, Nash et al. [2] conducted a study of respondents from a large registry of siblings based on a latent genetic risk factor with anxiety disorders, major depression, and neuroticism. Following the research, they obtained modest results of the correlations between serotonin system genes and anxiety. A study on animal models, conducted by Landgraf et al. [3], allowed the identification of genes responsible for anxiety as a trait, thus making the individual vulnerable to developing anxious comorbidity. Boehm et al. [4], following research on animals, concluded that the results of the study provide strong evidence that sensitivity to the effects of ethanol on locomotor behavior, anxiety-like behavior, and the stress axis have some genetic influence.

The human major histocompatibility complex (MHC) is located on the short arm of chromosome 6. Structurally and functionally, this region is the largest and most polymorphic in the human genome, representing ~1% of it. The first antigens encoded by these genes were discovered on the surface of leukocytes, so the locus also became known as the human leukocyte antigen (HLA) region. This region consists of two main classes: class I MHC molecules (HLA-A, B, and C) and class II MHC molecules (HLA-DR, DQ, and DP) [5]. In mammals, the MHC class I and MHC class II gene clusters are separated by a sequence stretch of about 700 kb called the MHC class III region [6].

Over the years, there have been numerous studies on the associations between different HLA alleles and infectious or immune diseases [7,8,9,10,11], but few studies have tried to find associations between HLA genotypes and different behaviors or mental states. For example, associations were found between HLA alleles and suicidal behavior [12,13], between HLA genotypes and aggression [14], and between the HLA system and psychosis [15] or panic disorder [16].

The present study began from theoretical approaches regarding genetic vulnerabilities in an individual related to anxiety traits [17], organic anxiety [18], and anxiety styles or profiles [19]. This approach focuses mostly on the role of the mediator (from a genetic point of view) in anxiety disorders and less on the cognitive schemas that can give intensity and psychopathological valence to the anxiety disorder. It started from the hypothesis that there are certain genes that can influence the response of people subjected to a stressful situation. Depending on the genetic profile, this response can be different: a low, medium, or high level of distress. The aim of the study was to determine some correlations between the HLA profile (class I and class II HLA genes) and the affective distress profile (PDA) in a certain stressful situation. In the present study, the data were collected during participation in a paternity test.

## 2. Materials and Methods

### 2.1. Participants

Between May 2019 and December 2020, 115 people between the ages of 22 and 61 participated in this study, with the average age being 38 years, 50.4% of the participants being female and 49.6% male. The groups met the criterion of homogeneity, presenting normal distributions, both from the perspective of age (skewness = 0.253 ≤ ±1.96; kurtosis = 0.443 ≤ ±1.96) and from the perspective of gender (skewness = 0.018 ≤ ±1.96; kurtosis = 1.703 ≤ ±1.96). All persons were participants in paternity tests at the Institute of Forensic Medicine in Cluj-Napoca, being the mothers of the children who were to be tested, respectively, and the alleged fathers.

The selection of research participants was on a voluntary basis, from the sample of patients examined at the Institute of Forensic Medicine. Before the beginning of the research, procedures were explained to all the participants, specifying the right to withdraw from the study at any time and requesting their voluntarily written informed consent. The study was approved by the Ethics Committee of the University of Medicine and Pharmacy in Cluj-Napoca (no. 272/16.06.2017).

### 2.2. Instruments

In order to determine the affective distress, the PDA test was used, the participants completing the questionnaires regarding the emotional distress of the last two weeks before the test. This tool for psychological evaluation and testing of anxiety presents a scale with 39 items that measure functional negative emotions from the categories “worry/anxiety” and “sadness/depression”, as well as positive emotions [20]. The test allows the calculation of a general affective distress score, a negative emotions score, a positive emotions score, as well as separate scores for “worry” (functional), “anxiety” (dysfunctional), “sadness” (functional), and “depression” (dysfunctional). The efficiency of the tool is enhanced by the fact that it allows both the estimation of a global anxiety value and the calculation of separate scores for functional and dysfunctional negative emotions, as well as for “worry” (functional), “anxiety” (dysfunctional), sadness (functional), and “depression” (dysfunctional). Compared to other tests for the assessment of anxiety, the PDA contains a relatively small number of items designed in an accessible language, being easy to manage and rate.

Cronbach’s Alpha correlations between PDA scores and other similar instruments: total score with other similar test batteries that achieved a significant threshold of *p* < 0.5* was: BDI .56* N173; DAS-A .26* N539; ATQ .48* 27* N330; ABS2-1B -.26* N326; ABS2-RB .74* N106; POMS - ALL .75* N110; POMS- NEG -.46* N112; POMS-POZ .58* N115; STAY - X1.47* N98; STAY- X2-.19* N115; USAQ .40* N108; YSQ-L2-.58* N115.

Cronbach’s Alpha correlations between scores on the scales of the PDA test were:

The score obtained on the PDA-sadness test correlations with a significant threshold of *p* < 0.5* was: .50* 174 .21* .555 .43* 117 .23* 344 -.21* 340 .70* 108 .72* 112 -.43* 115 .55* 118 .43* 101 -.15 118 .32* 111.

The score obtained on the PDA-worry scale correlations with a significant threshold of *p* < 0.5* was: .40* 176 .19* 558 .40* 118 .18* 348 -.21 344 .70* 110 .71* 114 - .44* 116 .59” 119 .42* 102 -.20* 119 .41” 111. 

The score obtained on the correlations in the PDA-functional negative emotions test with a significant threshold of *p* < 0.5* was: .50* 174 .22* 551 .44* 116 .22* 342 -.23* 338 .73* 108 .75 * 112 -.45* 114 .60* 117 .44* 100 -.18* 117 .37” ‘ 110.

The score obtained on the PDA-dysfunctional negative emotions scale correlations with a significant threshold of *p* < 0.5* was: -.57* 174 .28* 548 .49* 116 .30* 336 -.21′ 334 .71* 107 .72* 111 -.45” 114 .54* 117 .48* 100 -.18* 117 .43* 110.

The score obtained on the PDA-anxiety scale correlations with a significant threshold of *p* < 0.5* was: .50* 175 .25* 554 .46* 117 .28* 342 -.25* 338 .67* 108 .68* 112 -.42* 115 .55* 118 .44* 101 -.14 118 .43* 110.

The score obtained on the PDA-depression scale correlations with a significant threshold of *p* < 0.5* was: .57* 175 .21′ 555 .49* 118 .30* 342 -.26* 340 .70* 109 .71* 113 -.45* 116 .51” 119 .48* 102 -.20* 119 .41* 112.

### 2.3. DNA Extraction

From each person, 2 mL of peripheral venous blood was collected, and DNA was extracted using the Maxwell RSC Whole Blood DNA Kit (Promega Corporation, Madison, WI, USA), according to the manufacturer’s instructions. A Pearl Nanophotometer (Implen GmbH, Munich, Germany) was used in determining the DNA concentration and purity.

### 2.4. Identification of HLA Alleles

In 87 people, 5 genes were analyzed (HLA-A, HLA-B, HLA-C from HLA class I, respectively, HLA-DRB1 and HLA-DQB1 from HLA class II); in 16 people, only HLA genes from class I were determined (HLA-A, HLA-B, HLA-C), and in 12 people, only those from class II (HLA-DRB1 and HLA-DQB1). For the identification of the low-resolution HLA-A, HLA-B, HLA-C, HLA-DRB1, and HLA-DQB1 alleles, an HLA-FluoGene ABC kit (Inno-train Diagnostik GmbH, Kronberg, Germany) and an HLA-FluoGene DRDQ kit (Inno-train Diagnostik GmbH) were used, according to the manufacturer’s instructions, based on the Sequence Specific Priming Polymerase Chain Reaction (SSP-PCR). DNA amplification was carried out on a G-Storm thermal cycler (Gene Technologies Ltd., Essex, UK) and the mixture containing the extracted DNA sample was submitted to 40 amplification cycles (15 s at 96 °C and 60 s at 60 °C) after an initial denaturation step for 2 min at 95 °C. Detection of the PCR products was performed by measuring fluorescence signals on a FluoVista Analyzer (Inno-train Diagnostik GmbH, Kronberg, Germany), the endpoint fluorescence of the various fluorochromes before and after PCR being automatically calculated by the FluoGene analysis software.

### 2.5. Statistical Analysis

The database was constructed using the IBM SPSS Statistics, version 23.0 (IBM Corp., New York, SUA). The association between PDA, on one hand, and age, respectively, sex, on the other hand was analyzed. A correlation study was also conducted aiming to assess the potential relationships between the HLA profile and PDA. Univariate analyses were performed for the assessment of associations between the categorical variables using the Chi-squared test (χ2) and Fisher’s exact test. The Bonferroni correction (Adjusted α) was applied to *p*-values [21,22,23].

## 3. Results

### 3.1. Affective Distress Profile (PDA)

Depending on the scores obtained after applying the scale, people were divided into five categories: very low level of distress (≤28), low level of distress (29–39), medium level of distress (40–56), high level of distress (57–86), and very high level of distress (≥87).

A significant positive correlation was obtained between the PDA and age (r = 0.229; *p* = 0.014), indicating that as the individual advances in age, affective distress tends to increase.

The results regarding the PDA according to sex are presented in Figure 1.

### 3.2. Correlations between the HLA Profile and PDA

To determine the correlations between the PDA and the HLA profile, the subjects were divided into two categories: low and very low level of distress (≤39) marked with a low PDA, and a medium, high, and very high level of distress (≥40) marked with a medium and high PDA.

All 105 possible HLA-A, 219 HLA-B, 78 HLA-C, 66 HLA-DRB1, and 28 possible HLA-DQB1 types in pairs of alleles were described in absolute and relative frequencies for the two groups. Univariate analyses were performed for the assessment of associations between the categorical variables using the Chi-squared test (χ2), Fisher’s exact test, and Bonferroni correction (Adj α). The univariate analysis considered all 15 HLA-A, 23 HLA-B, 13 HLA-C, 12 HLA-DRB1, and 8 HLA-DQB1 alleles in a contingency table, determining the presence or absence of each allele in the low PDA group and medium and high PDA group.

The results for the univariate analysis are presented in Table 1, Table 2, Table 3, Table 4 and Table 5.

The contingency table indicates that the highest percentages are for HLA-A*01 in the low PDA group, for A*02 in both groups, A*03 in the medium and high PDA group, and A*32 also in the medium and high PDA group.

The univariate analysis revealed that the HLA-A*01 allele was associated to the low PDA group with a very highly significant *p* = 0.001 (Table 1), respectively, significant for A*30 (*p* = 0.048). HLA-A*32 (*p* = 0.028) was significantly associated to the medium and high PDA group. If the PDA and HLA-A alleles are compared using the Bonferroni correction, the *p*-value is 0.05/15 = 0.003333 to be significant. A *p*-value lower than 0.0033 was recorded only for the HLA-A*01 allele.

The highest percentages are for HLA-B*08 in the low PDA group, for B*35 in both groups, B*40 in the medium and high PDA group, and B*44 also in the medium and high PDA group.

The univariate analysis using SPSS showed that the HLA-B*08 allele (*p* = 0.022) was associated to the low PDA group with a significant *p* ≤ 0.05 and the HLA-B*52 allele (*p* = 0.018) was associated to the medium and high PDA group with a significant *p* ≤ 0.05 (Table 2).

The contingency table indicates high percentages for HLA-C*04, C*06, and C*07 in the low PDA group and C*12 in the medium and high PDA group.

The univariate data analysis showed that the HLA-C*12 allele (*p* = 0.015) was significantly associated to the medium and high PDA group at a level of significance of *p* ≤ 0.05 (Table 3).

The highest percentages are for HLA-DRB1*04 in both groups, the low PDA, and also in the medium and high PDA group and for the DRB1*11 in the low PDA group.

The univariate analysis revealed that the HLA-DRB1*11 allele (*p* = 0.005) was associated to the low PDA group at a level of significance of *p* ≤ 0.05, highly significant, (Table 4).

Very high percentages were obtained for HLA-DQB1*03 for both groups, the one with a low PDA and also the one with a medium and high PDA; for DQB1*05, also for both groups; and DQB1*06 in the medium and high PDA group.

## 4. Discussion

In the current study, the PDA was analyzed in 115 individuals who participated in paternity tests. After applying the scale, it was observed that there were people in all five categories, with most of the participants falling in the low and medium level of distress (Figure 1). The results obtained could be explained in terms of the individual’s ability to perceive the implications and responsibilities that may arise from taking such a test. Thus, the positive correlation obtained between the PDA and age indicates that this factor can have a significant influence on the emotional distress felt in the case of a paternity test. As the elderly individual advances, he tends to perceive the circumstantial particularities much more analytically, being aware of the implications and responsibilities he must assume.

Regarding the distribution according to sex, it was observed that in the very high level of distress category (≥87), there were only men. This association suggests that men feel the emotional distress more strongly in the case of this paternity test, probably through the lens of the uncertainty regarding the result, the mother always being the known one.

Among the 115 individuals, the HLA alleles from class I (HLA-A, HLA-B, and HLA-C) were determined for 103 people and those from class II (HLA-DRB1 and HLA-DQB1) for 99. To determine the alleles that correlate with the PDA, the participants were divided into two groups: a group with a low level of distress (including the very low and low level) and one with a medium and high level (including the medium, high, and very high level).

The results regarding the HLA-A gene showed that the A*02 allele was the most frequent in both groups in percentages over 50%. This result agrees with another study of ours regarding the frequency of alleles, carried out on 2794 people from the same region where the participants of the current study come from (Transylvania, Romania), in which the A*02 allele was shown to be much more frequent than the other HLA-A alleles [24]. The same was observed for the B*35 allele, frequent in both groups, being the most frequent HLA-B allele in the population of this region. This information regarding HLA profiles in populations can be particularly useful in studies on ancestral groups, as well as in the identification of rare functional mutations in the MHC system in individual populations, which may be associated with different pathologies or mental disorders.

The univariate analysis using SPSS showed that there are statistically significant correlations with four analyzed genes. In the case of individuals with emotional distress at a medium or high level, the presence of alleles -HLA-A*32, HLA-B*52, and HLA-C*12 was predominantly observed, which could indicate that individuals who are carriers of these alleles have a higher risk of developing anxiety disorders. On the other hand, carriers of alleles HLA-A*01, and A*30, HLA-B*08, and HLA-DRB1*11 do not seem to present this risk. The A*01 allele was associated to the low PDA group with a very highly significant *p* = 0.001 (Table 1), even after the Bonferroni correction.

This study is the first to explore the presence of possible correlations between the HLA profile and PDA. Although studies have generally focused on HLA associations with infectious or autoimmune diseases, there are few studies that have determined correlations between the HLA profile and mental or emotional disorders [12,16,25]. 

Some studies support some genetic influence in certain mental disorders, without identifying the genes involved. For example, it has been shown that there is some evidence for genetic effects on normal fear and anxiety, on anxiety disorders as a whole, and on panic and obsessive-compulsive disorders, agoraphobia, and blood-injury phobia [26]. Moreover, a study of 2000 showed that there is a genetic influence that can partially explain the risk of the concurrent occurrence of generalized anxiety disorder and panic disorder throughout life [25]. However, some authors claim that although panic disorder, generalized anxiety disorder, phobias, and obsessive-compulsive disorder have a significant familial aggregation, the hereditary transmission of anxiety disorders is in a modest range (30–40%), significantly lower than for disorders such as schizophrenia and bipolar disorder, with this hypothesis being largely explained by individual environmental factors [27].

Other studies have tried to find correlations between different genes and depression, anxiety, or stress. For example, a 2019 study that used gene networks, constructed by adding genes functionally related to genes from an initial set, suggested that there are hypothetical mechanisms for the realization of a “contentious” phenotype and these genes may be characterized by an ambiguous phenotypic manifestation. The study was based on the principle of an ambiguous phenotypic manifestation for the same polymorphisms in people from different populations or people exposed to different environmental influences [28]. Another study, conducted on rats and mice, demonstrated that the behavioral phenotype has been found to be significantly correlated with the expression of the neuropeptide arginine vasopressin (AVP) at the level of the hypothalamic paraventricular nucleus and contribute to hyper-anxiety and depression-like behavior and single-nucleotide polymorphisms (SNPs) in the regulation structures of the AVP gene represent the basis of the phenotypic phenomena [3].

Genome-wide association studies (GWAS) have been described in the literature, which identified genetic variants associated with depression. For example, the CONVERGE Consortium identified two genetic associations focusing on a sample of Chinese women with recurrent severe depression: two SNPs on chromosome 10 showed evidence of association, one near the SIRT1 gene and the other in an intron of LHPP [29]. In another study, conducted on over 300,000 individuals, 15 loci associated with depression were identified using a self-report of a clinical diagnosis [30]. Some researchers have tried to determine the genetic basis of panic disorder [31], and some suggestive associations have been observed in several loci, such as BDKRB2 or NPY5R, with this disorder [32]. An association between polymorphisms in the TMEM132D region and panic disorder was also reported in a Japanese population [16]. Genome-wide association studies (GWAS) for panic disorder focused on candidate SNPs with statistical significance were attempted, but a number of polymorphisms were determined that did not reach the threshold for genome-wide significance due to low allele frequencies, even if they were really involved in the pathogenesis [33].

Regarding the HLA system, studies to date have reported negative correlations between HLA-C*07 and schizophrenia [34,35] and positive correlations between HLA-B*58:01, C*07:01, DQA1*01:01, DQB1*05:01, and DPB1*17:01 and posttraumatic stress disorders [36]. In our study, the C*07 allele was present in a much higher percentage (42.19%) in the group with a low PDA compared to that with a medium and high PDA (28.21%), that is, it can be said that it shows a negative correlation with an increased risk of developing anxiety disorders. Another example of associations with the HLA system is a case report suggesting that the HLA-DRB1*03 allele may explain a common etiology underlying the comorbidity of Graves’ disease, type 2 diabetes, and schizophrenia in one patient [37].

The GWAS literature supports the involvement of the MHC locus in schizophrenia susceptibility, and current evidence suggests that the MHC plays a more significant role in schizophrenia susceptibility than in other psychiatric disorders [15].

Debnath et al. [38] analyzed the distribution of polymorphisms at the HLA-G locus in patients with bipolar disorder and found that the HLA-G 14 bp Ins/Ins genotype was significantly more prevalent in healthy controls than in patients and that the prevalence of such a protective genotype is more small among patients born during the winter season, compared to those born in other periods, but the possible mechanisms to explain the low expression of HLA-G and the susceptibility to bipolar disorder are still debatable.

HLA-B and HLA-DRB1 were investigated as candidate susceptibility genes for panic disorder in 744 subjects and 1418 control subjects. The study provided the initial evidence that the HLA-DRB1*13:02 allele is associated with panic disorder, but without clarifying the mechanisms that determine this [33]. In our case, the DRB1*13 allele was not associated with any of the two groups, the frequencies being appropriate (15.25 vs. 15.00%), so it cannot be said that it has any influence on anxiety disorders. The profile of chronic emotional distress can deepen the psychological or organic comorbidity. Our previous study revealed the fact that an individual with a psychological vulnerability can outline a severe affective profile, with significant cognitive-behavioral consequences depending on the result of the paternity test [39].

The present study addressed the implications of the HLA profile regarding the manifestation of anxiety disorders. Possible correlations between the HLA class I and II alleles and affective distress were identified. These results indicate that genes can have an influence on affective distress; in this sense, a potential genetic predisposition to the development of anxiety disorders can be estimated. These results can be a premise for a new direction of research regarding the genetic influence on anxiety disorders, given that there are studies that attest to the fact that immune system dysfunctions are one of the main mediators of psychiatric disorders [40].

Associations in the MHC system can be explained by the interactions between genes and the environment, a fact demonstrated by other diseases associated with HLA polymorphisms. The mechanisms that can explain the role of MHC in the different response to stress can be related to immunity, the gene expression of HLA alleles being related to immune and neuronal activity. Following the body’s exposure to a stressful factor, the activation of the immune system is triggered as a natural fighting response of the body, and depending on the body, the immune response is different. The immunogenetic baggage of an individual has an important role in modulating the response to stress, and one of the main elements of the immunogenetic baggage is the HLA locus, which is very polymorphic.

Our study also had certain limitations. It can be said that a relatively small number of subjects were studied, but this is compensated by the fact that a large number of gene loci were analyzed, five HLA loci. In addition, we had statistically significant results; the A*01 allele was associated to the low PDA group at a very highly significant level. On the other hand, we did not try to determine the genetic mechanisms by which the alleles can be associated with the stress response mode, but this was not our objective; we wanted to see whether or not there are some correlations between the HLA profile and the PDA.

Our study is a preliminary study, aiming to identify some HLA alleles that can influence the degree of anxiety of different subjects. We propose that in future studies on this topic, we determine certain parameters such as molecular tests for the common stress markers, for instance, markers of oxidative stress. The next challenge is to establish the molecular mechanisms by which gene loci mediate effects so that they can then be used as therapeutic targets. It is necessary to identify the genetic variants for anxiety or depression, so that in the future these discoveries can be used as clinical tools.

## 5. Conclusions

The present study highlighted the potential associations between HLA alleles and anxiety disorders, representing a possible premise for identifying the genetic peculiarities involved in the individual’s predisposition to anxiety disorders. The outcomes were also validated by the appropriate values of the statistical parameters. As an initial study investigating the effects of HLA allelic variations in anxiety disorders, the current study may have a broad impact in psychiatric genetics and immunology.

## Figures and Tables

**Figure 1 ijerph-19-12608-f001:**
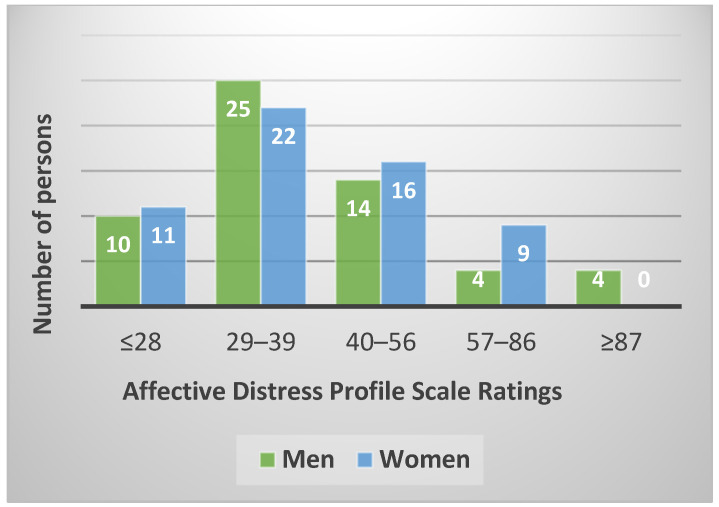
PDA by category depending on sex. ≤28: very low level of distress, 29–39: low level of distress, 40–56: medium level of distress, 57–86: high level of distress, ≥87: very high level of distress.

**Table 1 ijerph-19-12608-t001:** Correlation between PDA and HLA-A.

HLA-A (*n* = 103)	Low PDA (*n* = 64)		Medium + High PDA (*n* = 39)		χ2	*p*-Value	Adj α
	*n*	%	*n*	%			
**A*01**	**26**	**40.63**	**4**	**10.26**	**10.8265**	**0.0010**	0.003333
A*02	36	56.25	21	53.85	0.0567	0.8119	
A*03	10	15.63	9	23.08	0.8945	0.3443	
A*11	7	1094	2	5.13	1.0256	0.3112	
A*23	6	9.38	2	5.13	0.6101	0.4348	
A*24	9	14.06	3	7.69	0.9554	0.3284	
A*25	6	9.38	1	2.56	1.7747	0.1828	
A*26	9	14.06	5	12.82	0.0318	0.8584	
A*29	4	6.25	6	15.38	2.3066	0.1288	
**A*30**	**6**	**9.38**	**0**	**0.00**	**3.8824**	**0.0488**	
A*31	1	1.56	2	5.13	1.0896	0.2966	
**A*32**	**4**	**6.25**	**8**	**20.51**	**4.7893**	**0.0286**	
A*33	1	1.56	2	5.13	1.0896	0.2966	
A*68	4	6.25	3	7.69	0.0796	0.7779	
A*69	0	0.00	1	2.56	1.6571	0.1980	

Bold text indicates a statistically significant difference between the two groups.

**Table 2 ijerph-19-12608-t002:** Correlation between PDA and HLA-B.

HLA-B (*n* = 103)	Low PDA (*n* = 64)		Medium + High PDA (*n* = 39)		χ2	*p*-Value	Adj α
	*n*	%	*n*	%			
B*07	8	12.50	6	15.38	0.1717	0.6786	0.002173
**B*08**	**14**	**21.88**	**2**	**5.13**	**5.1797**	**0.0229**	
B*13	3	4.69	1	2.56	0.2927	0.5885	
B*14	1	1.56	3	7.69	2.4394	0.1183	
B*15	7	10.94	2	5.13	1.0256	0.3112	
B*18	9	14.06	6	15.38	0.0340	0.8536	
B*27	7	10.94	4	10.26	0.0118	0.9136	
B*35	19	29.69	10	25.64	0.1962	0.6578	
B*37	2	3.13	0	0.00	1.2429	0.2649	
B*38	3	4.69	3	7.69	0.3988	0.5277	
B*39	5	7.81	0	0.00	3.2023	0.0735	
B*40	7	10.94	7	17.95	1.0143	0.3139	
B*41	2	3.13	1	2.56	0.0270	0.8696	
B*44	9	14.06	7	17.95	0.2789	0.5974	
B*45	0	0.00	1	2.56	1.6571	0.1980	
B*47	0	0.00	2	5.13	3.3470	0.0673	
B*49	3	4.69	3	7.69	0.3988	0.5277	
B*50	1	1.56	1	2.56	0.1277	0.7209	
B*51	10	15.63	5	12.82	0.1532	0.6955	
**B*52**	**1**	**1.56**	**5**	**12.82**	**5.5986**	**0.0180**	
B*55	1	1.56	1	2.56	0.1277	0.7209	
B*56	3	4.69	0	0.00	1.8830	0.1700	
B*57	8	12.50	3	7.69	0.5872	0.4435	

Bold text indicates a statistically significant difference between the two groups.

**Table 3 ijerph-19-12608-t003:** Correlation between PDA and HLA-C.

HLA-C (*n* = 103)	Low PDA (*n* = 64)		Medium + High PDA (*n* = 39)		χ2	*p*-Value	Adj α
	*n*	%	*n*	%			
C*01	9	14.06	3	7.69	0.9554	0.3284	0.003846
C*02	7	10.94	7	17.95	1.0143	0.3139	
C*03	6	9.38	5	12.82	0.3016	0.5829	
C*04	20	31.25	9	23.08	0.8002	0.3710	
C*05	1	1.56	3	7.69	2.4394	0.1183	
C*06	19	29.69	6	15.38	2.6971	0.1005	
C*07	27	42.19	11	28.21	2.0349	0.1537	
C*08	3	4.69	3	7.69	0.3988	0.5277	
**C*12**	**11**	**17.19**	**15**	**38.46**	**5.8119**	**0.0159**	
C*14	0	0.00	1	2.56	1.6571	0.1980	
C*15	8	12.50	3	7.69	0.5872	0.4435	
C*16	3	4.69	2	5.13	0.0102	0.9196	
C*17	1	1.56	1	2.56	0.1277	0.7209	

Bold text indicates a statistically significant difference between the two groups.

**Table 4 ijerph-19-12608-t004:** Correlation between PDA and HLA-DRB1.

HLA-DRB1 (*n* = 99)	Low PDA (*n* = 59)		Medium + High PDA (*n* = 40)		χ2	*p*-Value	Adj α
	*n*	%	*n*	%			
DRB1*01	12	20.34	7	17.50	0.1239	0.7249	0.004166
DRB1*03	12	20.34	6	15.00	0.4568	0.4991	
DRB1*04	15	25.42	11	27.50	0.0531	0.8178	
DRB* 05	0	0.00	1	2.50	1.4901	0.2222	
DRB1*07	8	13.56	8	20.00	0.7298	0.3929	
DRB1*08	3	5.08	3	7.50	0.2443	0.6212	
DRB1*10	1	1.69	1	2.50	0.0781	0.7799	
**DRB1*11**	**28**	**47.46**	**8**	**20.00**	**7.7666**	**0.0053**	
DRB1*13	9	15.25	6	15.00	0.0012	0.9724	
DRB1*14	5	8.47	7	17.50	1.8230	0.1770	
DRB1*15	6	10.17	6	15.00	0.5222	0.4699	
DRB1*16	10	16.95	7	17.50	0.0051	0.9431	

Bold text indicates a statistically significant difference between the two groups.

**Table 5 ijerph-19-12608-t005:** Correlation between PDA and HLA-DQB1.

HLA-DQB1 (*n* = 99)	Low PDA (*n* = 59)		Medium + High PDA (*n* = 40)		χ2	*p*-Value	Adj α
	*n*	%	*n*	%			
DQB1*01	0	0.00	1	2.50	1.4901	0.2222	
DQB1*02	16	27.12	11	27.50	0.0017	0.9667	0.00625
DQB1*03	41	69.49	21	52.50	2.9405	0.0864	
DQB1*04	3	5.08	3	7.50	0.2443	0.6212	
DQB1*05	28	47.46	21	52.50	0.2425	0.6224	
DQB1*06	12	20.34	13	32.50	1.8677	0.1717	
DQB1*11	0	0.00	1	2.50	1.4901	0.2222	
DQB1*15	1	1.69	0	0.00	0.6849	0.4079	

## Data Availability

Data are contained within the article.

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
