# Peer review of "The Influence of HLA Alleles on the Affective Distress Profile"

_ijerph, 2022, doi:10.3390/ijerph191912608_

Round 1
Reviewer 1 Report
Please, see the detailed doc. file.
Thank you very much.

Reviewer 2 Report
In their pilot study, Vică and colleagues investigated an association between the affective distress profile (PDA) and the allelic polymorphism of the HLA loci. The HLA are seldom investigated in behavioral studies. This fact makes the manuscript potentially interesting and novel.
The study cohort was formed by certainly stressed healthy subjects who had undergone an affective distress due to personal circumstances such as passing forensic paternity tests. The authors revealed multiple associations, which can be interesting for the scientific community.
Unfortunately, the study design contains several serious defects. First and the most, the authors deemed "significant" each comparison with a P-value less than 0.20. The classic threshold is 0.05, and many experts declare a necessity to reduce it down to 0.01 or even less. I have never faced a research where hte threshold P-value exceeded 0.05. Secondly, the authors conducted multiple comparisons in a limited group, but did not apply the Bonferroni correction. The p-values must be re-calculated using Bonferroni and the results must be re-estimated correspondingly.
The other shortcomings of the manuscript cannot be corrected in the current situation with the data already available, so I'm writing them just for future studies. The sample size should be enlarged by an order of magnitude. The parameters studied should include data of molecular tests for the common stress markers, such as markers of oxidative stress , cfDNA, hormonal profiles, etc.
The last but not least is the absense of the control group of healthy subjects who is not stressed (no legal action, forensic testing and disease currently).
Finally, the Discussion should contain the minimum discourse of possible mechanisms underpinning the associations found.
Round 2
Reviewer 1 Report
Ok for publishing. Thank you very much for giving the chance of revise this manuscript
Reviewer 2 Report
The edited version is much better. The only correction is a stylistic amendment in lines 382-383: "We propose that in future studies on this topic we determine certain parameters such as molecular tests for the common stress markers, such as markers of oxidative stress."
It's not good to use "such as" twice in the same phrase. I'd advice to replace for "namely, markers of oxidative stress" or "for instance, markers of oxidative stress"
